# Causal relationships between lipid and glycemic levels in an Indian population: A bidirectional Mendelian randomization approach

**Tripti Agarwal**[1], **Tanica Lyngdoh**[1], **Frank Dudbridge**[2], **Giriraj Ratan Chandak**[3], **Sanjay Kinra**[4], **Dorairaj Prabhakaran**[5], **K. Srinath Reddy**[5], **Caroline L. Relton**[6], **George Davey Smith**[6], **Shah Ebrahim**[4], **Vipin Gupta**[7]*, **Gagandeep Kaur Walia**[5]*

**1** Indian Institute of Public Health-Delhi, Public Health Foundation of India, Gurgaon, India, **2** Department of Health Sciences, University of Leicester, Leicester, United Kingdom, **3** CSIR-Centre for Cellular and Molecular Biology, Hyderabad, India, **4** Department of Non-Communicable Disease Epidemiology, London School of Hygiene and Tropical Medicine, London, United Kingdom, **5** Public Health Foundation of India, Gurgaon, India, **6** MRC Integrative Epidemiology Unit, Bristol Medical School, University of Bristol, Bristol, United Kingdom, **7** Department of Anthropology, University of Delhi, Delhi, India

* gkaurw@gmail.com, gagandeep.k.walia@phfi.org (GKW); udaiig@gmail.com (VG)

**Data Availability Statement:** The data underlying the results presented in the study are available with the Indian Migration Study website hoted on

## Abstract

### Background

Dyslipidemia and abnormal glycemic traits are leading causes of morbidity and mortality. Although the association between the two traits is well established, there still exists a gap in the evidence for the direction of causality.

### Objective

This study aimed to examine the direction of the causal relationship between lipids and glycemic traits in an Indian population using bidirectional Mendelian randomization (BMR).

### Methods

The BMR analysis was conducted on 4900 individuals (2450 sib-pairs) from the Indian Migration Study. Instrument variables were generated for each lipid and glycemic trait (fasting insulin, fasting glucose, HOMA-IR, HOMA-β, LDL-cholesterol, HDL-cholesterol, total cholesterol and triglycerides) to examine the causal relationship by applying two-stage least squares (2SLS) regression in both directions.

### Results

Lipid and glycemic traits were found to be associated observationally, however, results from 2SLS showed that only triglycerides, defined by weighted genetic risk score (wGRS) of 3 SNPs (rs662799 at *APOAV*, rs780094 at *GCKR* and rs4420638 at *APOE/C1/C4*), were observed to be causally effecting 1.15% variation in HOMA-IR (SE = 0.22, P = 0.010), 1.53% in HOMA-β (SE = 0.21, P = 0.001) and 1.18% in fasting insulin (SE = 0.23, P =

London School of Hygiene and Tropical Medicine, UK server (URL: https://apcaps.lshtm.ac.uk/related-studies/ims/).

**Funding:** The Indian Migration Study was funded by the Wellcome Trust (grant number GR070797MF to Prof. Shah Ebrahim, LSHTM, UK). The genetic data generation was funded by a project grant from the Wellcome Trust (083541/Z/07/Z to Prof. Shah Ebrahim, LSHTM, UK). The present analyses were supported by Wellcome Trust/DBT India Alliance (grant no. IA/CPHE/16/1/502649 to Dr. Gagandeep Kaur Walia, PHFI). The funders were not involved in study design, data collection, analyses, manuscript preparation or decision for submission.

**Competing interests:** The authors have declared that no competing interests exist.

0.009). No evidence for a causal effect was observed in the reverse direction or between any other lipid and glycemic traits.

## Conclusion

The study findings suggest that triglycerides may causally impact various glycemic traits. However, the findings need to be replicated in larger studies.

## Introduction

Circulating lipids and glycemic traits are established risk factors for cardiometabolic disorders, a leading cause of morbidity and mortality. Dyslipidemia refers to high levels of triglycerides, total cholesterol, Low Density Lipoprotein-Cholesterol (LDL-C) and low levels of High Density Lipoprotein-Cholesterol (HDL-C). The prevalence of at least one of these conditions in India is as high as 79% [1]. An abnormal glycemic profile can indicate an insulin resistant state, pancreatic-β cell dysfunction, and imbalance in insulin and glucose levels. The prevalence of abnormal glycemic traits in India ranges from 11.2% to 12.8% [2, 3]. Both traits are highly heritable, ranging from 28% to 78% for serum lipid traits [4] and 10% to 75% for glycemic traits [5]. It has recently been hypothesized that the Homeostasis Model Assessment–Insulin Resistance (HOMA-IR) level is an important risk factor for dyslipidemia based on a study on 108 Turkish children [6]. Insulin plays an important role in altering lipids by inducing the synthesis, activation of lipoprotein lipase (LPL) and lipogenic enzymes which regulate the transport and metabolism of triglycerides [7–9]. On the contrary, there also exists evidence that triglyceride synthesis is independent of insulin resistance, insulin action or the levels of insulin [10].

The role of triglycerides in dysregulation of glucose metabolism and other glycemic traits has been extensively examined recently [11]. Reduced HDL-C and increased triglycerides have been reported to associated with an increase in fasting insulin secretion over 3.5 years in an European cohort [12]. Similarly, stimulatory action of excess triglycerides on β-cell release of insulin was observed in a multiethnic cohort of adolescents [13]. However, no such association between triglycerides and β-cell activity was observed in a retrospective study in a study on Italian growth data [14], however they did find HDL-C to be inversely associated with β-cell insulin secretion. Similar findings from a Chinese adult cohort, strongly suggests that dyslipidemia leads to insulin resistance (IR) mediated through altered insulin levels [15]. Although, dyslipidemia is known to be adversely associated with the mediators of T2D, the genetic risk of elevated triglycerides has been reported to be protective for Type 2 Diabetes in some studies [16–18]. Animal models support the role of inhibition of a key lipolytic enzyme, adipose triglyceride lipase (ATGL) in affecting insulin resistance [19, 20]. Further, Lipid biomolecules have demonstrated the potential to alter the activity of pancreatic-β cells, hence, insulin levels and also insulin signaling pathways [21–28].

Though, the observational association between lipids and glycemic levels is inevitable [6, 11, 12], the direction of the causal relationship between the two is still debatable due to inherent weaknesses of observational studies such as confounding and reverse causation. Moreover, the evidences suggest mechanistic pathways also in both the directions [7, 9, 19, 20]. Establishing the direction of causality is of global public health importance to improve the understanding of the patho-physiology of the cardiometabolic diseases and provide a deeper clinical

insight. This is further more required in the developing countries with huge unaddressed burden of cardiometabolic disorders and unique manifestations.

Thus, with the aim of identifying the direction of causality between the lipid and glycemic levels, a widely used approach akin to randomized controlled trials (RCTs), based on genetic variants, was undertaken on sample from the Indian Migration Study (IMS). A bidirectional Mendelian randomization (BMR) approach, based on natural randomization of genes at the time of birth, independent of confounders and other environmental factors, employs instruments derived from genetic variants as proxies for an exposure in the causal pathway, thereby addressing the problems of reverse causality and confounding. Thus, the present study was designed to understand the nature and direction of causality between lipids and glycemic traits among Indians using BMR to assess if increase/decrease in lipids lead to imbalance in glycemic traits or if alteration in lipid levels are a consequence of increase/decrease in glycemic traits.

## Methods

### Study population

The IMS is a cross-sectional population based sib-pair study [29, 30], with the participants recruited during the year 2005–2007 from factories located in four Indian cities—Lucknow, Nagpur, Hyderabad and Bangalore (S1 Appendix). The IMS was ethically approved by the All India Institute of Medical Sciences (AIIMS), New Delhi (reference number A-60/4/8/2004). All the all methods were performed in accordance with the national ICMR guidelines and regulations. Pre-informed written consent was obtained from each participant to use their de-identified phenotypic and genotypic data for research purposes in the future. The genetic study on which the current analyses is based, was ethically approved by London School of Hygiene and Tropical Medicine, UK (Ref. No. 5276) and Centre for Cellular and Molecular Biology, India (Ref. No. IEC/CCMB28/2008/6th Feb 2008). For the given analysis, a total of 4900 individuals (2450 sib-pairs) were included, from an initial 7067 participants, after excluding singletons, diabetics and CVD participants and those who had poor genotype data (Fig 1).

### Study procedures & measurements

Phenotyping details are described in the S1 Appendix. Fasting serum samples were used for generating data on fasting insulin and lipid levels including HDL-C, total cholesterol and triglycerides [30]. Fasting glucose was measured in fluoride plasma samples in local labs [30]. Estimation of LDL-C was done using the Friedewald-Fredrickson formula [31]. HOMA scores (HOMA-IR and HOMA-β, measures for insulin resistance and pancreatic β-cell activity respectively) were calculated using standard equations [32]. Data on diet, physical activity, alcohol consumption and tobacco smoking were recorded using interviewer-administered questionnaires [30].

### Genotyping and SNP selection

The candidate single nucleotide polymorphisms (SNPs) related to lipid and glycemic levels were genotyped on IMS participants using Sequenom Mass ARRAY during 2009–2010, based on the knowledge available at that time (see S1 Appendix for details). The detailed explanation of the genotyping method has been described previously [33, 34].

For the given study, data for total of 44 SNPs were available; 35 for glycemic traits [33] and 9 for lipid traits [34] in the IMS sample. Deviation from Hardy-Weinberg equilibrium (HWE) was tested using exact test on unrelated participants (only factory workers and spouses, sibs were excluded), in the overall IMS sample, as well as stratified for study sites. Any SNP that deviated from HWE (p<0.001 after Sidak correction) was excluded from the analysis.

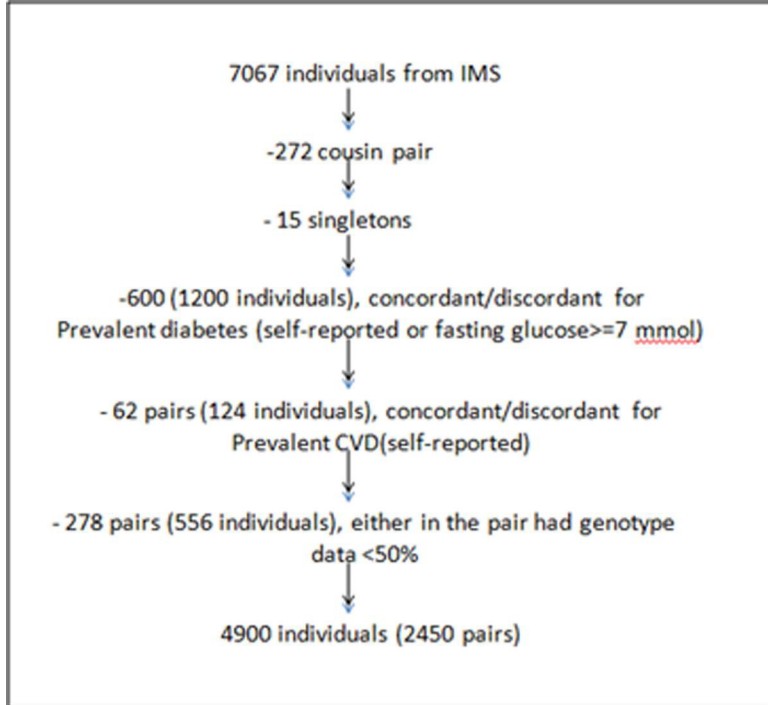

**Fig 1. Sampling strategy for present analyses from Indian Migration Study.**

The study was >80%powered to detect a Quantitative Trait Loci (QTL) explaining upt o 1% variation of a trait as derived using the Genetic Power Calculator using the "QTL association for sibships and singletons" [35], given the sample size of 2450 sib-pairs, α = 0.05 and Minor Allele Frequency (MAF) ranging from 1% to 45% (Table C in S1 Appendix). Using the mRnd Power calculator for MR analyses, the present study was65–68% powered to study 1–1.5% variation in the outcome trait, with instrument explaining 1% variation in the exposure among 2450 sib-pairs at α = 0.05.

## Statistical analyses

All of the analyses were performed in STATA v13.1. The quantitative traits were appropriately transformed depending upon the distribution. HOMA-IR, HOMA-β, fasting insulin and triglycerides, not being normally distributed, were log-transformed. All the variables were further standardized to facilitate comparability between coefficients.

As a preliminary step, the observational association between the lipid and glycemic traits was tested in both the directions using mixed linear model (to account for clustering due to family-effects) adjusted for age, sex, study site location and BMI (forced confounders). Further, the regression models were adjusted for dietary factors including daily energy intake, daily carbohydrate intake, daily fat intake; physical activity [Metabolic Equivalent of Task (MET)hrs/day]; alcohol consumption and tobacco smoking.

The genetic variants for lipid and glycemic levels were validated using adjusted mixed linear model, after accounting for family effects using an orthogonal family-based method described by Fulker [36]. Efforts were made to robustly check the three MR assumptions– 1. Genetic variants (SNPs) should be associated with the exposure of interest, 2. SNPs should be associated with the outcome, conditional on the exposure only and 3. SNPs should not be associated with

the confounders (see S1 Appendix for details). The SNPs that met all the assumptions were used for generating weighted genetic score(wGRS) using the equation shown below (1), where **w** and **a** represents the weight or the effect size and number of risk alleles respectively. The wGRS were then tested for association with the traits using the orthogonal Fulker method [36] and checked for all the MR assumptions.

$$Weighted\ Genetic\ Risk\ Score\ (wGRS) = (w1\ x\ a1) + \ldots\ldots + (wi\ x\ ai) \tag{1}$$

BMR was performed by Instrument Variable (IV) analysis using the generated instruments (wGRS) wherein the 2 Stage Least Square (2SLS) method was employed (Fig 2). This was done using *xtivregress* in Stata (*xt* was used to account for clustering due to family effects).

As in the association analyses, the mixed models were adjusted for age, sex, site and location and BMI. Further, adjustments for daily fat intake, daily carbohydrate intake, daily energy intake, physical activity (MET hrs/day), alcohol consumption and smoking were done in the second model. Endogeneity of the instruments was tested using the Durbin-Hausman test in which the estimates from the Ordinary Least Square (OLS) method were compared to that of 2SLS method using *hausman* command in Stata.

Finally, sensitivity analyses using different instruments as combinations of SNPs and individual SNPs was performed between the exposures and outcomes that were observed to be causally associated in IV analyses. Additionally, the SNPs showing a statistically significant causal association were tested for exclusion restriction criteria using inverse-variance weighted method of MR-Egger using the *mregger* command in Stata.

## Results

The basic demographic and clinical characteristics of all the 4900 study participants (2450 sib-pairs) are described in Table 1. The mean age of the participants is 39years. 43.04% individuals were females and overall 62.35% were urban residents. Males reported a comparatively higher mean for most of the risk-factors (Table 1). Tables A and B in S1 Appendix shows the observational association between lipid and glycemic traits in the IMS where associations were seen in both directions.

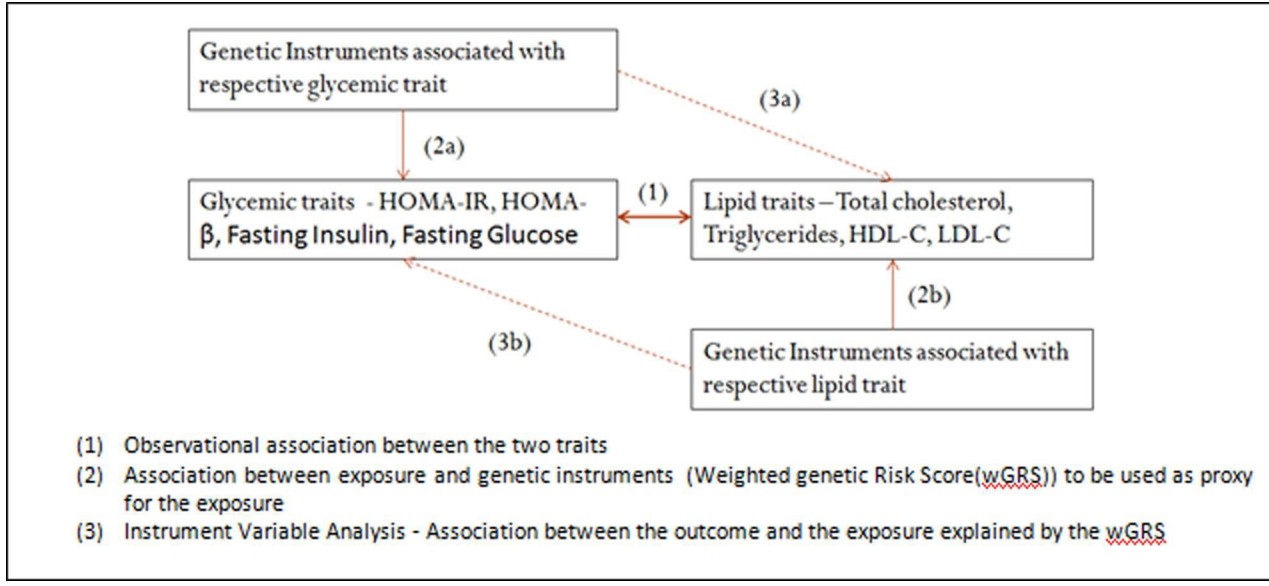

**Fig 2. Pictorial explanation of the application of Bidirectional Mendelian Randomization in the present analyses.**

**Table 1. Baseline characteristics of the IMS participants.**

| Characteristics | Total (N = 4900) | Male (N = 2791) | Female (N = 2109) | p-value[*] |
|---|---|---|---|---|
| | | | | |
| Age (Years) [$] | 39.48 (10.26) | 40.08 (10.55) | 38.68 (9.80) | 0.7212 |
| Gender [†] | | | | |
| Male | 2791 (56.96) | - | - | |
| Female | 2109 (43.04) | - | - | |
| Location [†] | | | | <0.001 |
| Urban | 3055 (62.35) | 1560 (55.89) | 1495 (70.89) | |
| Rural | 1845 (37.65) | 1231 (44.11) | 614 (29.11) | |
| Site [†] | | | | <0.001 |
| Bangalore | 844 (17.22) | 450 (16.12) | 394 (18.68) | |
| Hyderabad | 1290 (26.33) | 643 (23.04) | 647 (30.68) | |
| Nagpur | 1362 (27.80) | 858 (30.74) | 504 (23.90) | |
| Lucknow | 1404 (28.65) | 840 (30.10) | 564 (26.74) | |
| Fasting Glucose (mmol/l)[$] | 4.99 (0.64) | 5.013 (0.66) | 4.97 (0.62) | 0.0369 |
| Fasting Insulin [$] (mmol/l) [‡] | 1.66 (0.86) | 1.63 (0.88) | 1.70 (0.84) | 0.0073 |
| HOMA-IR[$‡] | 0.15 (0.89) | 0.12 (0.90) | 0.18 (0.86) | 0.019 |
| HOMA-β[$‡] | 4.36 (0.98) | 4.32 (1.01) | 4.40 (0.93) | 0.0035 |
| Total cholesterol (mmol/l)[$] | 4.67 (1.13) | 4.63 (1.13) | 4.73 (1.13) | 0.0013 |
| Triglycerides[$](mmol/l) [‡] | 0.25 (0.44) | 0.28 (0.45) | 0.20 (0.43) | <0.001 |
| HDL cholesterol (mmol/l)[$] | 1.17 (0.25) | 1.15 (0.25) | 1.20 (0.25) | <0.001 |
| LDL cholesterol (mmol/l)[$] | 2.86 (0.99) | 2.81 (0.99) | 2.92 (1.00) | 0.0001 |
| BMI (kg/m$^2$)[$] | 23.45 (4.46) | 22.88 (3.92) | 24.20 (4.99) | <0.001 |
| Total Physical Activity/ day (MET h/day)[$] | 38.97 (4.65) | 39.76 (4.89) | 37.93 (4.07) | <0.001 |
| Average daily carbohydrate intake (g/day)[$] | 460.11 (155.93) | 499.82 (164.11) | 407.57 (126.55) | <0.001 |
| Average daily fat intake (g/day)[$] | 84.03 (36.36) | 90.58 (38.86) | 75.37 (30.70) | <0.001 |
| Average daily energy intake (kcal/day)[$] | 2929.18 (1005.81) | 3182.74 (1055.15) | 2593.617 (824.48) | <0.001 |
| Alcohol consumption (Yes)[†] | 758 (15.47) | 676 (24.22) | 82 (3.89) | <0.001 |
| Smoker (Yes)[†] | 490 (10) | 485 (17.38) | 5 (0.24) | <0.001 |

[$] All continuous variables are reported as Mean (SD) and [†]categorical as n (%)

[*]Test of comparison (t-test and χ$^2$ test depending upon variable) between groups, p<0.05 signifies the groups are different for the variable

[‡]Geometric Mean

The measures are adjusted for clustering due to sibling effect; there are missing values for variables except age, gender, site and location

Out of 35 variants, five glycemic related SNPs deviated from HWE (Table C in S1 Appendix) and were excluded from the analysis. Therefore, 30 SNPs for glycemic traits and all 9 SNPs for lipid traits were included in the causal analysis. We could validate 7 SNPs for lipid levels (Table D in S1 Appendix) and 5 SNPs for glycemic traits (Table E in S1 Appendix). These validated SNPs were used to generate wGRS after testing for MR assumptions for exclusion restriction (Tables F—G in S1 Appendix) and association with confounders (Table H in S1 Appendix).

Two separate instruments were generated for total cholesterol using variants at *APOAV*, *APOB* and *LDLR* loci. As one of the variants for total cholesterol in *APOAV* was found to be associated with HOMA-β (violating MR assumption of exclusion restriction), a separate instrument using only *APOB* and *LDLR* variants was used for examining its causal effect of total cholesterol on HOMA-β. The wGRS was generated for triglyceride using variants at *APOAV*, *GCKR* and *APOE/C1/C4* and for LDL-C using variants at *APOB* and *LDLR* loci. No

**Table 2. Association of lipid traits (explained by instruments) with glycemic traits: Mendelian Randomization Analyses.**

| Instruments | | HOMA-IR | | HOMA-β | | Fasting Insulin | | Fasting Glucose | |
|---|---|---|---|---|---|---|---|---|---|
| | | β (SE) | p-value | β (SE) | p-value | β (SE) | p-value | β (SE) | p-value |
| TC (rs662799 + rs562338# + rs6511720$) | Model¹ | 0.28 (0.23) | 0.235 | † | † | 0.30 (0.24) | 0.21 | -0.08 (0.20) | 0.699 |
| | Model² | 0.21 (0.27) | 0.426 | † | † | 0.23 (0.26) | 0.382 | -0.09 (0.22) | 0.699 |
| TC (rs562338# + rs6511720$) | Model¹ | 0.18 (0.27) | 0.514 | 0.00 (0.28) | 0.991 | 0.16 (0.28) | 0.556 | 0.12 (0.25) | 0.637 |
| | Model² | 0.11 (0.31) | 0.720 | -0.01 (0.21) | 0.955 | 0.10 (0.31) | 0.751 | 0.09 (0.28) | 0.750 |
| TG (rs662799 + rs780094 + rs4420638) | Model¹ | 0.57 (0.22) | 0.010* | 0.69 (0.21) | 0.001* | 0.60 (0.23) | 0.009* | -0.08 (0.20) | 0.710 |
| | Model² | 0.56 (0.21) | 0.006* | 0.66 (0.22) | 0.002* | 0.58 (0.21) | 0.005* | -0.02 (0.17) | 0.917 |
| LDL-C(rs562338# + rs6511720$) | Model¹ | 0.17 (0.23) | 0.468 | 0.00 (0.23) | 0.997 | 0.15 (0.23) | 0.516 | 0.12 (0.23) | 0.604 |
| | Model² | 0.11 (0.27) | 0.677 | -0.04 (0.26) | 0.875 | 0.10 (0.27) | 0.721 | 0.10 (0.26) | 0.715 |

TC- Total Cholesterol, TG–Triglycerides

Model¹—Adjusted for Age, sex, site, location and BMI

Model²—Adjusted for Age, sex, site, location, BMI, average daily fat intake, average daily energy intake, average daily carbohydrate intake, MET Score/day, alcohol consumption and smoking

β (SE)–Standardized coefficients i.e SD unit change in outcome per SD unit increase in exposure (Standard Error)

†SNP/Instrument associated with the outcome, even after adjusting for exposure, thus excluded from analyses as it did not meet MR assumption

#rs562338$rs6511720 were associated with Daily Fat intake and physical activity respectively, which were used as confounders in Model 2

*Level of significance p<0.05

valid instrument could be generated for HDL-C as none of the SNPs met all the assumptions. For glycemic traits, wGRS combining variants at *ADAM30* and *CDKN2A* was used as an instrument for HOMA-IR and fasting insulin. wGRS generated by combining variants at *CDKAL1* and *TCF7L2* were used as an instrument for fasting glucose. A single SNP at *ADAM30* associated with HOMA-β was used as its instrument.

The instruments generated were significantly associated with their respective traits, cumulatively showing greater effect (Table I in S1 Appendix). The instruments were also tested for associations with their outcome and confounders to rule out pleiotropic effects (Table I in S1 Appendix). All the instruments followed all the 3 MR assumptions (Table I in S1 Appendix) and were thus used for IV analyses.

The results of the 2SLS method of IV analysis are reported in Tables 2 and 3. Only triglycerides showed positive causal effects on different glycemic traits, except fasting glucose (Table 2). Every 1% increase in triglyceride levels were observed to be causally associated with 1.15% variation in HOMA-IR score (β = 0.57, SE = 0.22, p = 0.010), 1.53% variation in HOMA-β score (β = 0.69, SE = 0.21, p = 0.001) and 1.18% variation in fasting insulin level (β = 0.60, SE = 0.23, p = 0.009). Even after additionally adjusting for lifestyle factors like diet, physical activity, alcohol consumption and smoking, significant causal associations were found (Table 2). Since association between lipid and glycemic traits is evident in observational analyses, we did not apply a multiple testing correction (MTC) in IV analyses. However, even after accounting for a MTC (level of significance = 0.003), a causal effect was evident between triglycerides and HOMA- β. The effect estimate from the 2SLS analysis was significantly less as compared to estimate from associations between phenotypic traits as seen from Durbin-Hausman test (Table J in S1 Appendix). However, since most of the p-values are less than 0.001 in 2SLS, it signifies that results from MR analyses and observational analyses (Durbin-Hausman test) are significantly different. No evidence could be generated favoring causal relationship between any other lipid and glycemic traits (Table 2). Further, no significant evidence was observed favoring causality in reverse direction with glycemic traits affecting lipid levels (Table 3).

**Table 3. Association of glycemic traits (explained by instruments) with lipids: Mendelian Randomization Analyses.**

| Instruments | | Total Cholesterol | | Triglycerides | | HDL-C | | LDL-C | |
|---|---|---|---|---|---|---|---|---|---|
| | | β (SE) | p-value | β (SE) | p-value | β (SE) | p-value | β (SE) | p-value |
| HOMA-IR (rs2641348 + rs10811661) | Model[1] | -0.33 (0.34) | 0.331 | 0.36 (0.32) | 0.267 | 0.20 (0.29) | 0.476 | † | † |
| | Model[2] | -0.30 (0.28) | 0.280 | 0.36 (0.31) | 0.242 | 0.22 (0.30) | 0.457 | † | † |
| HOMA-β (rs2641348) | Model[1] | -0.56 (0.62) | 0.376 | -0.14 (0.52) | 0.785 | 0.76 (0.57) | 0.185 | -0.86 (0.53) | 0.108 |
| | Model[2] | -0.60 (0.60) | 0.313 | -0.16 (0.56) | 0.772 | 0.83 (0.75) | 0.263 | -0.87 (0.57) | 0.128 |
| FI (rs2641348 + rs10811661) | Model[1] | -0.32 (0.33) | 0.326 | 0.33 (0.31) | 0.280 | 0.20 (0.28) | 0.462 | † | † |
| | Model[2] | -0.29 (0.26) | 0.264 | 0.34 (0.30) | 0.252 | 0.22 (0.29) | 0.437 | † | † |
| FG (rs7756992[#]+ rs7903146) | Model[1] | -013 (0.35) | 0.720 | 0.63 (0.43) | 0.146 | 0.43 (0.39) | 0.267 | -0.52 (0.37) | 0.160 |
| | [#]Model[2] | -0.17 (0.32) | 0.587 | 0.57 (0.40) | 0.160 | 0.42 (0.39) | 0.280 | -0.53 (0.45) | 0.243 |

FI- Fasting Insulin, FG- Fasting Glucose

Model[1]—Adjusted for Age, sex, site, location and BMI

Model[2]—Adjusted for Age, sex, site, location, BMI, average daily fat intake, average daily energy intake, average daily carbohydrate intake, MET Score/day, alcohol consumption and smoking

β (SE)–Standardized coefficients i.e SD unit change in outcome per SD unit increase in exposure (Standard Error)

†SNP/Instrument associated with the outcome, even after adjusting for exposure, thus excluded from analyses as it did not meet MR assumption

#rs7756992 was associated with smoking that was used as confounder in Model 2

The 3 SNPs used for score generation of triglycerides at *APOAV*, *GCKR* and *APOE/C1/C4*, that showed a causal effect on glycemic traits, were further exposed to sensitivity analysis. The wGRS of 3 SNPs were additionally tested for pleiotropy using the MR–Egger test. The significance level of p = 0.410, found in MR-Egger test, signifies zero-intercept using the *ivw* option (i.e. no directional pleiotropy); hence supporting the exclusion restriction assumption tested using the traditional method (Table F in S1 Appendix). IV analyses were performed using all possible combinations of the 3 triglyceride SNPs (Table 4) after checking these instruments for their independent association with outcomes and confounders (Table K in S1 Appendix). rs662799 at *APOAV* was observed to be significantly causally associated only with HOMA-β, individually (β = 0.56, SE = 0.18, p = 0.002) and also in combination with rs4420638 at *APOE/C1/C4* (β = 0.44, SE = 0.18, p = 0.015) (Table 4). Single SNP, rs4420638 at *APOE/C1/C4* did not show any significant causal effect individually with any of the glycemic trait. rs780094 at *GCKR* was associated with HOMA-IR (β = 0.90, SE = 0.43, p = 0.037), HOMA-β (β = 1.06, SE = 0.49, p = 0.031) and fasting insulin (β = 0.93, SE = 0.44, p = 0.034). The wGRS of rs780094 at *GCKR* and rs662799 at *APOAV* was found to be associated with HOMA-IR (β = 0.81, SE = 0.31, p = 0.010), but did not follow the assumption of exclusion restriction for HOMA-β and Fasting Insulin as the outcome (Table K in S1 Appendix). According to the sensitivity analyses, rs662799 at *APOAV* appears to be the most robust instrument for triglycerides for MR analyses.

## Discussion

It is extremely important to establish causality, independent of confounding and reverse causation, to fully understand the pathogenesis of cardiovascular disease and to inform effective interventions to prevent disease onset. BMR was applied to examine the direction of causality between lipid and glycemic traits in a general Indian population of mean age of ~40 yrs. Using conventional epidemiological methods, it was found that glycemic levels and lipid levels were observationally associated. However, owing to effect of confounders and reverse causation, no conclusion with regard to the direction of causation could be made from these associations.

**Table 4. MR sensitivity analyses for different combination of triglyceride SNPs with glycemic variables.**

| Instruments (*Loci*) | | HOMA-IR | | HOMA-β | | Fasting Insulin | | Fasting Glucose | |
|---|---|---|---|---|---|---|---|---|---|
| | | β (SE) | p-value | β (SE) | p-value | β (SE) | p-value | β (SE) | p-value |
| rs662799 | Model[1] | 0.30 (0.18) | 0.102 | 0.56 (0.18) | 0.002* | 0.35 (0.18) | 0.060 | -0.25 (0.22) | 0.247 |
| (*APOAV*) | Model[2] | 0.28 (0.23) | 0.224 | 0.51 (0.24) | 0.036* | 0.33 (0.23) | 0.151 | -0.23 (0.20) | 0.257 |
| rs780094 | Model[1] | 0.90 (0.43) | 0.037* | 1.06 (0.49) | 0.031* | 0.93 (0.44) | 0.034* | -0.05 (0.32) | 0.868 |
| (*GCKR*) | Model[2] | 1.01 (0.82) | 0.220 | 1.30 (0.933) | 0.163 | 1.05 (0.84) | 0.209 | -0.09 (0.46) | 0.840 |
| rs4420638 | Model[1] | -0.31 (0.50) | 0.537 | -0.53 (0.49) | 0.286 | -0.36 (0.51) | 0.485 | 0.32 (0.46) | 0.484 |
| (*APOE/C1/C4*) | Model[2] | -0.34 (0.52) | 0.507 | -0.63 (0.45) | 0.160 | -0.41 (0.50) | 0.414 | 0.40 (0.35) | 0.240 |
| rs662799 + rs780094 | Model[1] | 0.81 (0.31) | 0.010* | † | † | † | † | -0.21 (0.24) | 0.379 |
| (*APOAV + GCKR*) | Model[2] | 0.86 (0.35) | 0.013* | † | † | † | † | -0.19 (0.26) | 0.463 |
| rs662799 + rs4420638 | Model[1] | 0.39 (0.19) | 0.041* | 0.44 (0.18) | 0.015* | 0.04 (0.19) | 0.159 | 0.03 (0.21) | 0.894 |
| (*APOAV + APOE/C1/C4*) | Model[2] | 0.35 (0.19) | 0.061 | 0.37 (0.24) | 0.127 | 0.34 (0.19) | 0.064 | 0.11 (0.16) | 0.491 |
| rs780094 + rs4420638 | Model[1] | 0.30 (0.26) | 0.256 | 0.30 (0.27) | 0.262 | 0.29 (0.26) | 0.263 | 0.10 (0.29) | 0.731 |
| (*GCKR + APOE/C1/C4*) | Model[2] | 0.30 (0.32) | 0.349 | 0.26 (0.26) | 0.319 | 0.28 (0.29) | 0.333 | 0.15 (0.24) | 0.524 |
| rs662799 + rs780094 + rs4420638 | Model[1] | 0.57 (0.22) | 0.010* | 0.69 (0.21) | 0.001* | 0.60 (0.23) | 0.009* | -0.08 (0.20) | 0.710 |
| (*APOAV + GCKR + APOE/C1/C4*) | Model[2] | 0.56 (0.21) | 0.006* | 0.66 (0.22) | 0.002* | 0.58 (0.21) | 0.005* | -0.02 (0.17) | 0.917 |

Model[1]—Adjusted for Age, sex, site, location and BMI

Model[2]—Adjusted for Age, sex, site, location, BMI, average daily fat intake, average daily energy intake, average daily carbohydrate intake, MET Score/day, alcohol consumption and smoking

β (SE)–Standardized coefficients i.e SD unit change in outcome per SD unit increase in exposure

†SNP/Instrument associated with the outcome, even after adjusting for exposure, thus excluded from analyses as it did not meet MR assumption

*Level of significance p<0.05

Conversely, the results from IV analysis method provided evidence suggesting the causal role of triglycerides in increasing the different glycemic levels. No other lipid trait showed causal association with any of the glycemic trait, or in the reverse direction. Thus our findings suggest that an increase in triglyceride levels, explained by the related genetic instruments, casually impacts levels of HOMA-IR, HOMA-β and fasting insulin even after adjusting for all possible confounders.

Although the association between the lipid and glycemic traits has been established in India, not enough evidence has been generated to examine the causal relationship. Animal models indicate the role of increased triglycerides to have implications on insulin resistance (IR) through defective beta-oxidation and mitochondrial dysfunction [37]. Attempts from longitudinal studies worldwide have also been made to establish causality. Findings of the present analysis accord with the results from a Chinese cohort study of 3325 participants which reports that an increase in triglycerides precedes an increase in insulin and IR status [15]. Also, in a European cohort with 1016 non-diabetic volunteers of the age range 30–60 years (the characteristics of the cohort being similar to the sample in the present analysis), it was observed that an increase in triglycerides was associated with an increase in fasting insulin secretion over a period of 3.5 years, thereby indicating that triglycerides may play a causal role in changes in fasting insulin [12]. It was also observed in the Danish Inter99 cohort, which included 3,474 non-diabetic individuals, that over a period of 5 years a wGRS comprised of 39 genetic variants was associated with an increase in serum triglyceride levels and change in insulin resistance acted as an effect modifier. Although the results indicated that increased insulin resistance accentuated the effect of wGRS, no causality can be deduced from these findings [38].

Some European studies have showed inverse association between genetically explained triglycerides and risk of T2D [16, 17]. Association of reduced incidence of T2D with genetic risk

for elevated triglycerides was found in a MR study as well [18]. On the other hand, another MR Study conducted by De Silva et al among Europeans [39] did not report any association between circulating triglyceride levels and diabetes risk, fasting glucose, or fasting insulin. However, since the setting of the aforementioned European studies and the present study is different, it is important to consider the role of different environments leading to different gene-environment interaction. Moreover, the variants used for instrument generation are different and our primary outcomes were glycemic levels rather than type 2 diabetes.

Regarding other lipid traits, other than triglycerides, MR studies did not support the causal role of HDL-C in increasing the glucose levels and the risk of diabetes [40]. This was partially in line with the findings from the present study. The *HMGCR* gene, associated with lower LDL-C encoding for the target of statin (HMG-CoA reductase), has been reported to be causally associated with increased Type 2 Diabetes risk, higher plasma insulin and glucose [41]. This is also in concordance with the results from a RCT study where statin treatment was similarly associated with increased T2D risk hence supporting the causal role of lipids in contributing to an abnormal glycemic profile [41]. However, no causal role of LDL-C was observed in the present MR analysis where we used 2 SNPs in the LPL as instrumental variables. A recent review compiling the existing evidence from all the MR studies based on the same research question also suggests the inconclusive nature of the findings [42]. However, most studies aimed to study T2D as an outcome rather than studying the intermediate glycemic levels that were examined in present analyses.

Our study findings do not support a causal effect of increased glycemic levels on lipid levels, as seen in conventional observational methods. This could be due to the potential confounders/reverse causation leading to spurious associations in observational association analyses. This is supported by an animal study by Vatner et al showing that triglyceride synthesis and its levels are independent of insulin resistance, insulin action or the levels of insulin [10]. In contrast, it has been found from some animal studies that insulin affects lipid levels via synthesis and activation of *LPL*, an enzyme responsible for the removal of triglycerides from the plasma [7], or reduced expression of a key lipolytic enzyme [19, 20] or altered activities of the lipogenic enzymes [8]. The effect of insulin on *LPL* explains the association between glycemic traits and the variants at *LPL* after adjusting for triglycerides (Table 3), thereby leading to its elimination at the instrument generation stage. In the present study, *LPL*SNPs were found to be associated with glycemic trait independent of triglyceride and hence we removed it from the instrument generation.

The main strengths of the study are the multi-centric population-based sib-pair study design, and high quality genetic data. Despite the multiple advantages of MR, it also has certain limitations and thus care has to be taken to address these. First, we applied BMR which has the potential to give reliable evidence for causal directions since it employs instruments for both the dependent and independent variables [43]. Secondly, our sib-pair design is resistant to population stratification [43]. Thirdly, we generated wGRS using multiple SNPs that were used as instruments, which tend to increase the percentage variation of each trait explained thereby increasing the power of the study in comparison to use of a single SNP [43]. Even though the issue of pleiotropy cannot be ruled out completely, efforts have been taken to minimize its influence by including multiple variants as instrumental variables. Therefore, it is less likely that all variants in a wGRS will exhibit pleiotropy. Moreover, it is unlikely that all the variants of the wGRS will be in LD with the pleiotropic variant having association with the outcome or confounders. Fourth, it can also be stated that attempts have been made to maximize the validity of the instruments in the context of IV analysis. The validated SNPs used were robustly associated with the exposure in the present study. Each significant SNP was considered for instrument generation only if it was not independently associated with confounders

and/or the outcomes. MR Egger analysis was also performed to confirm the absence of directional pleiotropy in addition to routine methods. The generated wGRS instruments were further tested for association with confounders and outcomes (when adjusted for their respective exposure). The strength of the instruments could also be deduced from the F-statistics obtained (F-statistics ranging from 36.71 to 70.92) and the strength of association between the instrument and the trait is also very high. Fifth, the generalizability of the study is also increased due to the fact that the sample was taken from the IMS, which involves participants from north, south and central India. Also, to the best of our knowledge, this study is among few that have examined the causal association between glycemic and lipid traits.

The study also has its own limitations. The weights used for risk score generation were derived internally from the sample, which could produce biased estimates [44]. Secondly, although being one of the few MR studies in India with the largest known sample size, the study was still under-powered to detect more precise smaller estimates. Also, it is important to consider the translational effects of the variants which might further have an impact on the biological pathway.

Based on the study findings, it is evident that an increase in triglycerides may be causal in increasing the HOMA-scores and insulin levels. Hence, it may be but cautiously inferred that an increase in triglyceride levels can be a factor for impaired insulin activity and also affect the pancreatic- β cell activity. However, considering the limitations of the study, there is a need for a further larger study using more robust risk scores (generated using the more recently validated GWAS SNPs [45, 46–47] so that conclusions can be drawn reliably. Also, it is important to understand the function, transcriptional and translational pathways of the identified variants, including through the use of computational techniques and animal models, so that the findings can be biologically validated. Moreover, efforts can be taken to gather phenotypic and genotypic data in the same context from other Indian studies and execute another well-powered, pooled MR study. Studies can also be designed for different age groups, genders and ethnicities to capture effect potential modification. This is necessary to generate reliable causal evidence so that targeted clinical, and preventive interventions can be designed.

## Supporting information

**S1 Appendix.**
(DOCX)

## Acknowledgments

We are extremely thankful to Indian Migration Study team for facilitating this work.

## Author Contributions

**Conceptualization:** Tripti Agarwal, Tanica Lyngdoh, Sanjay Kinra, Dorairaj Prabhakaran, K. Srinath Reddy, George Davey Smith, Shah Ebrahim, Vipin Gupta, Gagandeep Kaur Walia.

**Data curation:** Tripti Agarwal, Giriraj Ratan Chandak, Sanjay Kinra, Vipin Gupta, Gagandeep Kaur Walia.

**Formal analysis:** Tripti Agarwal, Vipin Gupta, Gagandeep Kaur Walia.

**Funding acquisition:** Dorairaj Prabhakaran, K. Srinath Reddy, George Davey Smith, Shah Ebrahim, Vipin Gupta, Gagandeep Kaur Walia.

**Investigation:** Tanica Lyngdoh, Vipin Gupta, Gagandeep Kaur Walia.

**Methodology:** Tanica Lyngdoh, Frank Dudbridge, Caroline L. Relton, George Davey Smith, Shah Ebrahim, Vipin Gupta, Gagandeep Kaur Walia.

**Project administration:** Tripti Agarwal, Tanica Lyngdoh, Shah Ebrahim, Gagandeep Kaur Walia.

**Resources:** Frank Dudbridge, Giriraj Ratan Chandak, Sanjay Kinra, Dorairaj Prabhakaran, K. Srinath Reddy, George Davey Smith, Shah Ebrahim, Vipin Gupta, Gagandeep Kaur Walia.

**Software:** Tripti Agarwal, Frank Dudbridge, Gagandeep Kaur Walia.

**Supervision:** Tanica Lyngdoh, Dorairaj Prabhakaran, K. Srinath Reddy, Caroline L. Relton, George Davey Smith, Shah Ebrahim, Gagandeep Kaur Walia.

**Validation:** Tripti Agarwal, Vipin Gupta, Gagandeep Kaur Walia.

**Visualization:** Tripti Agarwal, Gagandeep Kaur Walia.

**Writing – original draft:** Tripti Agarwal, Gagandeep Kaur Walia.

**Writing – review & editing:** Tanica Lyngdoh, Frank Dudbridge, Giriraj Ratan Chandak, Sanjay Kinra, Dorairaj Prabhakaran, K. Srinath Reddy, Caroline L. Relton, George Davey Smith, Shah Ebrahim, Vipin Gupta, Gagandeep Kaur Walia.

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
