## [Decision Letter · Decision Letter 0]

30 Oct 2019

PONE-D-19-22133

Causal relationships between lipid and glycemic levels in an Indian population: A bidirectional Mendelian randomization approach

PLOS ONE

Dear Dr Walia,

Thank you for submitting your manuscript to PLOS ONE. After careful consideration, we feel that it has merit but does not fully meet PLOS ONE’s publication criteria as it currently stands. Therefore, we invite you to submit a revised version of the manuscript that addresses the points raised during the review process.

We would appreciate receiving your revised manuscript by Dec 14 2019 11:59PM. To enhance the reproducibility of your results, we recommend that if applicable you deposit your laboratory protocols in protocols.io, where a protocol can be assigned its own identifier (DOI) such that it can be cited independently in the future. For instructions see: http://journals.plos.org/plosone/s/submission-guidelines#loc-laboratory-protocols

We look forward to receiving your revised manuscript.

Kind regards,

Paolo Magni

Academic Editor

PLOS ONE

Journal Requirements:

1. Please include additional information regarding the survey or questionnaire used in the study and ensure that you have provided sufficient details that others could replicate the analyses. For instance, if you developed a questionnaire as part of this study and it is not under a copyright more restrictive than CC-BY, please include a copy, in both the original language and English, as Supporting Information.

Reviewers' comments:

Reviewer's Responses to Questions

**Comments to the Author**

1. Is the manuscript technically sound, and do the data support the conclusions?

Reviewer #1: Yes

2. Has the statistical analysis been performed appropriately and rigorously? 

Reviewer #1: Yes

3. Have the authors made all data underlying the findings in their manuscript fully available?

Reviewer #1: Yes

4. Is the manuscript presented in an intelligible fashion and written in standard English?

Reviewer #1: Yes

5. Review Comments to the Author

Reviewer #1: This interesting study by Agarwal et al. examined the causal relationships between plasma lipids and glycemic traits in 4,900 participants of the Indian Migration Study using bidirectional Mendelian randomization.

The authors identified 3 gene variants associated with circulating triglycerides and developed a weighted genetic risk score for triglycerides that correlated with fasting plasma insulin and with markers of insulin sensitivity and beta cell function. These findings support a causal impact of hypertriglyceridemia in the early alterations of glucose metabolism that characterize the pathogenesis of type 2 diabetes.

The question addressed by this trial is relevant from a clinical and physiological perspective. The effort to avoid potential confounders and limit the biases typical of MR studies is laudable and is a strength of the study. The manuscript is overall well written. Some issues and limitations listed below should be carefully considered, along with those already mentioned by the authors in the Discussion.

Major comments:

1. The lack of more accurate indexes of glucose tolerance, insulin sensitivity and beta cell function provided by dynamic metabolic tests is a major limitation of the study. The HOMA-b in particular is a poor marker of beta cell function for at least two reasons: it does not provide information on glucose-stimulated insulin secretion and it does not take into account potential differences in insulin clearance, which is known to be associated with plasma lipids and significantly affected by the ethnic background. The second issue might be circumvented using fasting levels of C peptide instead of insulin, if available.

2. The choice of the different confounders for lipids and glycemic traits is somehow questionable. It is not clear why the BMI was included in mixed linear models only for glycemic outcomes and not for lipids, since there is a clear correlation between body weight and plasma lipid profile. Moreover, triglyceride levels seem to be influenced more by the carbohydrate intake than by the fat intake (Samaha FF et al, NEJM 2003, doi: 10.1056/NEJMoa022637). Triglyceride and cholesterol levels are also influenced by the alcohol intake, which was available in the Indian Migration Study (Table 1).

3. The main finding of this study, which is the impact of triglycerides on glycemic traits and especially on beta cell function, has been extensively examined in recent years (Seghieri M et al., Diabetes Metab 2017, doi: 10.1016/j.diabet.2017.04.010). The authors should carefully revise the literature and discuss concordant and conflicting results.

The genetic susceptibility for higher triglycerides has been associated with a paradoxically lower risk for type 2 diabetes (Klimentidis YC et al, PLoS Genet 2015, doi: 10.1371/journal.pgen.1005204; White et al. JAMA Cardiol 2016, doi: 10.1001/jamacardio.2016.1884; Ahmad S et al, Arterioscler Thromb Vasc Biol 2019, doi: 10.1161/ATVBAHA.118.311562).

With regards to insulin secretion, circulating triglycerides have been reported to have a positive effect on beta cell function, regardless of the ethnic background, in some (Tricò D et al, Diabetes Obes Metab 2018, doi: 10.1111/dom.13467) but not all studies (Bedogni G et al, Endocr Connect 2019, 10.1530/EC-19-0333).

4. The number of SNPs available and eventually selected for the development of each genetic instrument was consistently lower compared with similar Mendelian randomization analyses (see for instance the above-mentioned studies).

5. At the end of the Discussion it is stated that “an increase in triglyceride levels can be a factor for […] pancreatic beta cell dysfunction”, which is apparently in contrast with the current study findings. Although I personally agree that, in the long term, insulin hypersecretion may lead to beta cell dysfunction, this study shows a positive rather than negative effect of triglyceride on the HOMA-b.

6. The Introduction is rather long and might benefit from being shortened and more focused. Appropriate references should be provided in the second paragraph.

7. The statement “The study was adequately powered…” is too vague. The authors should provide the actual power of the study.

Minor comment

1. I would change “p=0.000” with “p<0.001” in supplementary tables

6. PLOS authors have the option to publish the peer review history of their article (what does this mean?). If published, this will include your full peer review and any attached files.

Reviewer #1: No

---

## [Author Response · Author response to Decision Letter 0]

14 Dec 2019

Dear Editor

Thank you for informing your decision and sending the comments and suggestions from the reviewers which have helped in improving the manuscript.

We have responded to each comment below in blue coloured text and have revised the manuscript accordingly.

Reviewer #1: This interesting study by Agarwal et al. examined the causal relationships between plasma lipids and glycemic traits in 4,900 participants of the Indian Migration Study using bidirectional Mendelian randomization.

The authors identified 3 gene variants associated with circulating triglycerides and developed a weighted genetic risk score for triglycerides that correlated with fasting plasma insulin and with markers of insulin sensitivity and beta cell function. These findings support a causal impact of hypertriglyceridemia in the early alterations of glucose metabolism that characterize the pathogenesis of type 2 diabetes.

The question addressed by this trial is relevant from a clinical and physiological perspective. The effort to avoid potential confounders and limit the biases typical of MR studies is laudable and is a strength of the study. The manuscript is overall well written. Some issues and limitations listed below should be carefully considered, along with those already mentioned by the authors in the Discussion.

Response: We are thankful to the reviewer for recognizing the importance of examining causal relationships between cardiometabolic traits and dour efforts to address the possible limitations of Mendelian randomization (MR) design. We have now revised the manuscript according to the below mentioned comments and suggestions.

Major comments:

1. The lack of more accurate indexes of glucose tolerance, insulin sensitivity and beta cell function provided by dynamic metabolic tests is a major limitation of the study. The HOMA-b in particular is a poor marker of beta cell function for at least two reasons: it does not provide information on glucose-stimulated insulin secretion and it does not take into account potential differences in insulin clearance, which is known to be associated with plasma lipids and significantly affected by the ethnic background. The second issue might be circumvented using fasting levels of C peptide instead of insulin, if available.

Response: We agree with the reviewer that glycemic indices have evolved over time and currently more precise and clinically relevant markers are available. However, the present analyses were done utilizing the available de-identified data from Indian Migration Study which was conducted during 2005-2008 when only these glycemic levels (Fasting glucose, fasting insulin, HOMA-IR and HOMA-β) were generally collected in large epidemiological surveys.

2. The choice of the different confounders for lipids and glycemic traits is somehow questionable. It is not clear why the BMI was included in mixed linear models only for glycemic outcomes and not for lipids, since there is a clear correlation between body weight and plasma lipid profile. Moreover, triglyceride levels seem to be influenced more by the carbohydrate intake than by the fat intake (Samaha FF et al, NEJM 2003, doi: 10.1056/NEJMoa022637). Triglyceride and cholesterol levels are also influenced by the alcohol intake, which was available in the Indian Migration Study (Table 1).

Response: We would like to thank the reviewers to point out the other potential confounders that we missed in our analyses. As per the suggestion, we examined all the possible confounders that were available in the study and have now revised our analyses after including BMI uniformly in model-1 for both glycemic and lipid outcomes along with age, gender, location and site; and additional confounders like carbohydrates, alcohol use and tobacco smoking in model-2 along with previously included confounders for dietary factors and physical activity.

3. The main finding of this study, which is the impact of triglycerides on glycemic traits and especially on beta cell function, has been extensively examined in recent years (Seghieri M et al., Diabetes Metab 2017, doi: 10.1016/j.diabet.2017.04.010). The authors should carefully revise the literature and discuss concordant and conflicting results.

The genetic susceptibility for higher triglycerides has been associated with a paradoxically lower risk for type 2 diabetes (Klimentidis YC et al, PLoS Genet 2015, doi: 10.1371/journal.pgen.1005204; White et al. JAMA Cardiol 2016, doi: 10.1001/jamacardio.2016.1884; Ahmad S et al, Arterioscler Thromb Vasc Biol 2019, doi: 10.1161/ATVBAHA.118.311562).

With regards to insulin secretion, circulating triglycerides have been reported to have a positive effect on beta cell function, regardless of the ethnic background, in some (Tricò D et al, Diabetes Obes Metab 2018, doi: 10.1111/dom.13467) but not all studies (Bedogni G et al, Endocr Connect 2019, 10.1530/EC-19-0333).

Response: We would like to thank the reviewer for sharing the above studies. We have now included these studies along with other related recent literature in the introduction and discussion sections.

4. The number of SNPs available and eventually selected for the development of each genetic instrument was consistently lower compared with similar Mendelian randomization analyses (see for instance the above-mentioned studies).

Response: We agree with the reviewer that the recent MR studies utilize the allelic scores based on more number of variants compared to the present study. The present study was a small Masters thesis work based on available de-identified genotype and phenotype data. 

The genotype data on only 9 lipids and 35 glycemic loci was available in the study and out of these only those variants were included in the instrument that fulfilled Hardy-Weinberg Equilibrium and the three MR assumptions. We hope that the reviewers will agree that despite the lower number of variants, the genetic instruments were very robust and did provide very insightful findings. We would also like to state that there are very few MR studies from low resource settings like India and even those are based on smaller sample size and weaker instruments. However, we do understand that despite the various attempts, the present study has some limitations which have been discussed in the manuscript. Based on these preliminary results, our team is now involved in larger studied for examining causal relationships between various cardiometabolic traits utilizing more robust genetic instruments.

5. At the end of the Discussion it is stated that “an increase in triglyceride levels can be a factor for […] pancreatic beta cell dysfunction”, which is apparently in contrast with the current study findings. Although I personally agree that, in the long term, insulin hypersecretion may lead to beta cell dysfunction, this study shows a positive rather than negative effect of triglyceride on the HOMA-b.

Response: We would like to thank the reviewer for pointing this mistake. We have now revised the text accordingly.

6. The Introduction is rather long and might benefit from being shortened and more focused. Appropriate references should be provided in the second paragraph.

Response: As per reviewer suggestions, we have revised the introduction section and have provided relevant references.

7. The statement “The study was adequately powered…” is too vague. The authors should provide the actual power of the study.

Response: We have now provided the actual power of the study to examine the causal association using instrument variable regression in the methods section.

Minor comment

1. I would change “p=0.000” with “p<0.001” in supplementary tables

Response: We have revised the tables as suggested by the reviewer.

---

## [Decision Letter · Decision Letter 1]

13 Jan 2020

Causal relationships between lipid and glycemic levels in an Indian population: A bidirectional Mendelian randomization approach

PONE-D-19-22133R1

Dear Dr. Walia,

We are pleased to inform you that your manuscript has been judged scientifically suitable for publication and will be formally accepted for publication once it complies with all outstanding technical requirements.

With kind regards,

Paolo Magni

Academic Editor

PLOS ONE

Additional Editor Comments (optional):

Reviewers' comments:

Reviewer's Responses to Questions

**Comments to the Author**

1. If the authors have adequately addressed your comments raised in a previous round of review and you feel that this manuscript is now acceptable for publication, you may indicate that here to bypass the “Comments to the Author” section, enter your conflict of interest statement in the “Confidential to Editor” section, and submit your "Accept" recommendation.

Reviewer #1: All comments have been addressed

2. Is the manuscript technically sound, and do the data support the conclusions?

Reviewer #1: Yes

3. Has the statistical analysis been performed appropriately and rigorously? 

Reviewer #1: Yes

4. Have the authors made all data underlying the findings in their manuscript fully available?

Reviewer #1: Yes

5. Is the manuscript presented in an intelligible fashion and written in standard English?

Reviewer #1: Yes

6. Review Comments to the Author

Reviewer #1: The authors have addressed my comments in a satisfactory manner. I appreciate particularly the inclusion of additional potential confounders in the analyses, and the work put by the authors in their rebuttal and revisions. Please note that authors' first and last names are inverted in some of the new references. In my opinion, this issue may be addressed by the authors or the Editorial office after acceptance, without further revisions.

7. PLOS authors have the option to publish the peer review history of their article (what does this mean?). If published, this will include your full peer review and any attached files.

Reviewer #1: Yes: Domenico Trico

---

## [Editor Report · Acceptance letter]

21 Jan 2020

PONE-D-19-22133R1 

Causal relationships between lipid and glycemic levels in an Indian population: A bidirectional Mendelian randomization approach 

Dear Dr. Walia:

I am pleased to inform you that your manuscript has been deemed suitable for publication in PLOS ONE. Congratulations! Your manuscript is now with our production department. 

With kind regards,

on behalf of

Prof. Paolo Magni 

Academic Editor

PLOS ONE